# Canine Stem Cell Conditioned Media Accelerates Epithelial Migration in the Canine Tympanic Membrane

**DOI:** 10.3390/vetsci9020069

**Published:** 2022-02-06

**Authors:** Hyerin Suh, Suhyun Kim, Taeho Oh, Seulgi Bae

**Affiliations:** Department of Veterinary Internal Medicine, College of Veterinary Medicine, Kyungpook National University, 80 Daehak-ro, Daegu 41566, Korea; hysuh@knu.ac.kr (H.S.); itssummer93@knu.ac.kr (S.K.); thoh@knu.ac.kr (T.O.)

**Keywords:** dog, epithelial migration, stem cell, stem cell conditioned media, tympanic membrane

## Abstract

Similar to skin, epithelia in the tympanic membrane (TM) regenerate and move toward the opening of the external ear canal, a process called epithelial migration (EM). EM is important for maintaining healthy ears because this process removes cerumen and debris. Therefore, increasing the rate of EM or TM regeneration could be very important for healthy ear maintenance and function. Stem cells or their conditioned media have been used in medical therapy in humans to increase the rate and efficacy of EM. The purpose of this study was to evaluate the ability of canine stem cell conditioned media to accelerate EM in canine TMs. Canine adipose tissue derived-mesenchymal stem cell conditioned media (cAD-MSCCM), and several cytokines related to keratinocyte growth or migration within the media were quantified using ELISA. Ink drops were placed on the TMs of four normal beagles. Then, cAD-MSCCM was applied weekly, a total of three times to the TMs of one ear, and nothing was applied to the other eye. The results showed a higher TM EM rate in the treatment group than in the control group (*p* < 0.05). No adverse events were recorded. These results suggest that the weekly application of cAD-MSCCM accelerates the TM EM rate.

## 1. Introduction

The ear is involved in the detection and analysis of sound and the perception of the body’s position in its environment. It is anatomically and functionally divided into the external, middle, and inner ear [1]. While the external and middle ear are primarily involved in hearing, the inner ear’s role is mainly maintaining balance via the vestibular system [2]. The external acoustic meatus, which is the passageway that leads from the outside of the ear, receives sound waves and conducts them to the tympanic membrane (TM), which is also known as the eardrum [3]. The TM is an oval, thin tissue composed of three layers [4], and its external surface contains no hair, glands, or pigmentation. The TM is located between the external acoustic meatus and the middle ear, separating the tympanic cavity from the external ear [3]. Its outer layer is continuous with the cutaneous tissue on the external canal surface, its inner layer is continuous with the mucous membrane of the tympanic cavity, and its middle layer is a vascular connective tissue that gives the membrane its tension and stiffness [4].

Similar to skin, epithelia in the TM and external ear canal regenerate [5]. These renewing cells move radially from the TM toward the opening of the external ear canal [6,7,8,9,10]. This epithelial migration (EM) is a unique physiological process in the external ear that prevents the accumulation of desquamated substances, cerumen, and debris in the external acoustic meatus [11]. EM facilitates hearing, maintains a healthy canal environment, and repairs TM perforations [5]. Faster EM leads to a faster regeneration of the TM, resulting in the protection of the middle and inner ear from external infectious agents. Therefore, increasing the rate of EM or TM regeneration is very important for healthy ear maintenance and normal otic function.

Several studies have been conducted to identify the rate and pattern of EM [6,7,8,9,10]. Recently, stem cells or stem-cell-conditioned media (SCCM) have been used to increase the rate and efficacy of EM [12,13,14,15]. Stem cells are believed to enhance cutaneous wound healing through several processes [13]. Mesenchymal SCCM contains various growth factors and cytokines; hence, it may be considered as a better alternative to stem cell therapy [16]. It is also known to benefit wound healing and EM of the TM [14,15,16,17].

This study evaluates the ability of canine SCCM to accelerate EM of the TM. Canine adipose tissue-derived mesenchymal SCCM (cAD-MSCCM) was obtained, and several cytokines related to keratinocyte growth or migration were quantified in the conditioned media. Then, the rate of EM of TMs in normal beagle dogs was calculated using the ink drop method to evaluate the efficacy of the SCCM treatment.

## 2. Materials and Methods 

### 2.1. Collection of Canine Adipose Tissue-Derived Mesenchymal Stem Cells (cAD-MSCs)

Canine adipose tissue-derived mesenchymal stem cells (cAD-MSCs) were isolated from the inguinal fat of a beagle dog following the guidelines of the Institutional Animal Care and Use Committee of Kyungpook National University, Republic of Korea (IACUC: 2020-0081). A mixture of DMEM/F12 (HyClone laboratories, Logan, UT, USA), 10% fetal bovine serum (HyClone laboratories, Logan, UT, USA), and 1% gentamycin (Shin Poong Pharm., Seoul, Korea) was prepared in a T75 culture flask. Then, 10 ng/mL epidermal growth factor (PeproTech, Rocky Hill, NJ, USA), 10 ng/mL insulin-like growth factor (PeproTech, Rocky Hill, NJ, USA), and 2 ng/mL basic fibroblast growth factor (PeproTech) were added. cAD-MSCs were cultivated at a temperature of 37 °C and 5% CO_2_. When the cAD-MSCs reached the second passage, immunophenotyping was conducted by applying fluorescence-activated flow cytometry. PE-conjugated anti-CD21 antibody, PC5.5-conjugated anti-CD44 antibody, and APC-conjugated anti-CD90 antibody were used as positive markers. FITC-conjugated anti-CD34 antibody, FITC-conjugated anti-CD45 antibody, and FITC-conjugated anti-MHC2 antibody were used as negative markers. All antibodies used for acquisition were obtained from Bioscience, USA, CytoFLEX (Beckman Coulter, Brea, CA, USA) [18]. After 2–3 days, when confluency reached at least 70%, the CM was collected and then centrifuged at 300× *g* for 10 min. Using a 10 mL syringe, the supernatant was filtered through a sterile 0.2 μm filter membrane (Sigma-Aldrich, Sigma, St. Louis, MO, USA) to avoid cell fragment contamination. The subculture process and CM collection were repeated to ensure a sufficient supply. The collected CM was kept refrigerated at 5 °C until use.

### 2.2. Collection of Canine Keratinocyte-Conditioned Media (cKCM)

Canine progenitor epidermal keratinocytes (CELLnTEC Advanced Systems, Bern, Switzerland) were cultured in a T75 cell culture flask and CnT-09 medium (CELLnTEC Advanced Systems, Bern, Switzerland) at 37 °C and 5% CO_2_ until their confluency reached at least 70% [19]. Then, the subculture, CM collection, and storage method were repeated as for cAD-MSCCM.

### 2.3. Quantification of Basic Fibroblast Growth Factor (bFGF), Interleukin-6 (IL-6), and Vascular Endothelial Growth Factor-A (VEGF-A) in SCCM

Levels of canine basic fibroblast growth factor (bFGF), interkeukin-6 (IL-6), and vascular endothelial growth factor-A (VEGF-A) from cAD-MSCCM and cKCM were separately evaluated with ELISA kits (RayBiotech, Norcross, GA, USA) according to the manufacturer’s protocol. Optical density values were measured at a wavelength of 450 nm with a microplate spectrophotometer (Thermo Fisher Scientific, Waltham, MA, USA).

### 2.4. Laboratory Animals

Four two-year-old male beagles (ORIENT BIO, Seongnam, Korea) were used (IACUC: 2020-0123). A separate individual room was provided for each dog. The environment was kept at a room temperature of 24 °C and a relative humidity level of 50%. Food (Special Max, Sajo DongA Corporation, Chungnam, Dangjin, Korea) was provided once a day, and water was given ad libitum. During the research period, no oral medications were given except for heartworm preventives, and no topical preparations were applied. Physical examinations and bloodwork were performed within 3 months prior to the ink drop placement.

### 2.5. Evaluation of the Ear Canals

The ear canals were evaluated with handheld otoscopy (Welch Allyn, Skaneateles Falls, NY, USA) according to the Otitis Index Score (OTIS). Criteria such as erythema, edema, erosion/ulceration, exudate, and pain were measured using a score of 0 to 3, where higher scores represented greater severity [20]. In addition, any damage to the TM was evaluated. Ear swab samples were microscopically evaluated with a magnification power of 1000× to determine the presence of infection or inflammation. Dogs with a total OTIS above 4, ruptured TM, or infection in the ear canal were excluded from the study.

### 2.6. Ink Spotting, Repeated Re-Evaluation, and cAD-MSCCM Application

For general anesthesia, 0.04 mg/kg of medetomidine hydrochloride (Domitor^®^, Zoetis, Espoo, Finland) was injected intravenously, and 5 mg/kg of ketamine hydrochloride (Yuhan, Seoul, Korea) was injected intramuscularly. A customized instrument was developed with a 1 mL syringe and a PTFE Teflon tube with 0.5 mm inner diameter and 0.9 mm outer diameter (Uxcell, Hong Kong, China) (Figure 1). Waterproof tattoo ink (Tana Lee Temporary Coloring Tattoo Ink, Oyecoskorea, Incheon, Korea) was used. Using a video otoscope (Karl Storz Veterinary Endoscopy America, Goleta, CA, USA), an ink droplet was placed on the posterior quadrant of the pars tensa (PT). The day the ink drop settled on the TM was considered Day 0. The TM was re-evaluated by filming with the AIDA-vet video capture system (Karl Storz Veterinary Endoscopy America, Goleta, USA). Images showing the width of the malleus and the whole PT were captured. Ink drops were evaluated on Days 3 ± 1, 7 ± 1, and 14 ± 1, until either day 21 ± 1 or until the ink deviated from the PT. The image from day 0 was visualized on a laptop computer screen during re-evaluation [6]. Each beagle TM was divided into a control group or a treatment group. No substance was applied to the control group, while the treatment group had 0.05 mL of cAD-MSCCM applied directly on the TM every week using video otoscopy.

### 2.7. Malleus Width Measurement for Calibration

The malleus width was measured for calibration work for all TM EM rate evaluations. A piece of 3M index tape was cut into a 1.5 × 1.5 mm^2^ to use as a reference scale and then applied close to the short process of the malleus under general anesthesia (Figure 2). After image capture, all the scales were removed with a sterile saline flush. In Adobe Photoshop (version 22.5.1, 2021, Adobe systems, San Jose, CA, USA), the width of the malleus and the referential scale were measured in pixel units. Using the ratio to the reference scale, the width of the malleus was converted to micrometers.

### 2.8. Evaluation of the TM EM Rate

The captured TM images were aligned periodically with Photoshop. The center points of figures from two consecutive time intervals were connected with lines. All the distances were added and divided by the number of evaluation days to obtain the TM EM rate per day.

### 2.9. Statistical Analysis

For statistical analysis, Prism version 9.3.0 (GraphPad Software Inc., San Diego, CA, USA) was used. The levels of paracrine factors in cAD-MSCCM and cKCM were compared statistically using the Mann–Whitney test. The nonparametric paired *t*-test was used to compare the difference between the TM EM rate of the control group and that of the treatment group and to compare the width of the malleus of both ears. A *p*-value of < 0.05 was considered statistically significant.

## 3. Results 

### 3.1. Levels of bFGF, IL-6, and VEGF-A in cAD-MSCCM and cKCM

A comparison of the concentrations of bFGF, IL-6, and VEGF-A from cAD-MSCCM and cKCM is shown in Figure 3. The level of bFGF was significantly greater in cAD-MSCCM than in cKCM (*p* < 0.05). There was no significant difference in IL-6 expression between the groups (*p* > 0.05), and the level of VEGF-A expressed in cAD-MSCCM was significantly lower than that in cKCM (*p* < 0.05).

### 3.2. Malleus Width Measurement and Calibration

The average measured width of the malleus was 1064.0 ± 204.2 and 980.0 ± 26.8 μm for the left and right ears (LE and RE), respectively (Table 1). There was no significant difference in the width of the malleus between the LE and RE in any dog (*p* > 0.05).

### 3.3. Evaluation of the TM EM Rate and Effect of cAD-MSCCM

Eight different TMs (from four LEs and four REs) were evaluated twice, once as the control group and again as the treatment group. During management, the LE of beagle 3 suffered trauma; hence, its TM EM rate in the control group was excluded. The comparison of the ink movement between the two groups is shown in Figure 4. All ink drops moved outwards, but it was difficult to determine the direction of movement. The TM EM rate for each group is shown in Figure 5. The TM EM rate was significantly higher in the treatment group than in the control group, according to the paired *t*-test (*p* < 0.05). There was no significant difference between the TM EM rate of the LE and RE in the two groups. No adverse effect from cAD-MSCCM was found in the treatment group.

## 4. Discussion

In humans, the TM is known to repair itself very effectively, but failure in repair, which leads to chronic TM perforation, is seen in up to 15% of patients [17]. When this occurs, tympanoplasty is the treatment of choice [12]. Several studies have evaluated the use of stem cells and SCCM as alternative therapies [12,13,14,15]. Bone-marrow-derived stem cells applied as a treatment for a perforated TM have shown a high healing capacity [12]. Mesenchymal stem cells have also been shown to promote TM reconstruction in a rat model [13]. In another study, the use of CM from adipose stem cells led to human keratinocyte and fibroblast proliferation, which stimulated in vitro wound healing [14]. The canine TM heals itself effectively within 35 days under normal conditions [21]. In dogs, secondary TM rupture and otitis media (inflammation of the middle ear) occur in up to 50% of cases with prolonged otitis externa (inflammation of the outer ear) [22]. However, the exact prevalence of TM repair failure is not consistently reported, and neither surgical nor medicinal treatments are established in dogs.

The level of paracrine factors measured in an in vitro trial elucidated the basis for the phenomenon observed in vivo. In this study, the paracrine activity of canine keratinocytes (cKs) was used as the control for an in vitro trial because the outer layer of the TM is continuous with the cutaneous tissue on the external canal surface [1] and consists of keratinocytes [4]. Keratinocytes as well as fibroblasts are known to play important roles in cutaneous healing [14]. Epithelial proliferation and keratin movement toward the perforation are key changes related to TM repair [23]. Several cytokines and growth factors facilitate the recovery of a perforated TM through angiogenesis and connective tissue reconstruction [23]. VEGF-A, which is upregulated after wounding, works as a potential stimulus for angiogenesis [24]. bFGF and VEGF are well-known critical growth factors [23], and IL-6 is also known to stimulate the proliferation of keratinocytes [25]. In this study, bFGF, VEGF, and IL-6 were expressed in both cAD-MSCCM and cKCM. However, the bFGF level was significantly elevated in cAD-MSCCM compared with that in cKCM. cAD-MSCCM application provides additional paracrine factors from the exterior and may therefore have an additional stimulatory effect. The higher level of bFGF in cAD-MSCCM than in cKCM may suggest that bFGF is a relatively more stimulating factor than VEGF and IL-6 for TM EM acceleration. According to a previous study, bFGF stimulates keratinocyte migration [26] and improves the healing of perforated TMs when applied topically [27]. bFGF treatment has been shown to lead to hyperplasia near the malleus and annulus when directly applied to a perforated rat TM [28]. Epithelial hyperplasia around the annulus and malleus region is known to increase during the early stage of TM healing [29]. Accordingly, cAD-MSCCM may support the early stage of TM restoration. 

In this in vivo study, the weekly application of cAD-MSCCM improved the TM EM rate without adverse effects. As shown in Figure 5 and Table 2, the mean ± S.D. for the TM EM rates were 212.5 ± 66.4 μm/d and 133.1 ± 30.9 μm/d for the treatment and control groups, respectively. According to Tabacca et al. [6], the normal PT TM EM rate in dogs is 96.4 ± 43.1 μm/d; this finding is similar to or slower than the rate in the control group in this study. This might be due to aging, which slows down the TM EM rate [7]; some dogs in the Tabacca study were older than the ones involved in this study. In this study, the TM EM rate of the treatment group was clearly faster than in that in the Tabacca study. There was no significant difference between the TM EM of both ears. This was observed in both the treatment and control groups. The mean ± S.D. for the width of the mallei was 1022 ± 142.2 μm for two-year-old beagles. Tabacca’s study found that the average mallei width measured with a handheld caliper was 1.0 mm [6]. The width of the malleus is an essential indicator that allows TM EM measurement utilizing otoscopes. Unlike previous studies, the measurement method used in this study showed reliability without euthanasia, hence avoiding the need for animal sacrifice. 

cAD-MSCCM would be an effective alternative or ancillary therapy not only for chronic TM defects but also for other TM issues. In general, when there is clinical evidence of otitis media while the TM is intact, myringotomy is performed [21]. After flushing out purulent material inside the tympanic cavity, the early closure of a perforated TM might help reduce the risk of re-infection. Furthermore, this method may facilitate acute TM perforation recovery and would allow an earlier introduction of topical otic solutions that provoke inflammation when exposed to the middle ear [30]. EM is a protective and cleaning mechanism for the ear. cAD-MSCCM, by accelerating TM EM, might be suitable as an alternative therapy for otitis externa. To date, unlike the use of antibiotics, no resistance has been observed when using CM.

Considering animal welfare, only normal TMs were used in this study, and the growth factors evaluated correlated with wound healing. Clinical trials with client-owned dogs with perforated TMs should be performed. In clinics, topical agents for otitis are applied as often as twice daily. Additional study is needed to evaluate the safety and EM enhancement associated with increased application frequency. This study is a significant and meaningful step in that direction, given that it developed a method of evaluating the TM without sacrificing the animal subjects and demonstrated the efficacy of cAD-MSCCM to accelerate EM TM in dogs.

## 5. Conclusions

In this study, the weekly application of cAD-MSCCM in beagle dogs showed an accelerating effect on TM EM with no adverse effect in vivo. cAD-SMCCM seems to provide an additional paracrine factor from the exterior and additional stimulatory effect on TM. In vitro, among bFGF, IL-6, VEGF-A, the level of bFGF was higher in cAD-MSCCM when compared with cKCM, hence may be a relatively more stimulating factor than the others. cAD-MSCCM would be an effective alternative or ancillary therapy for TM issues.

## Figures and Tables

**Figure 1 vetsci-09-00069-f001:**
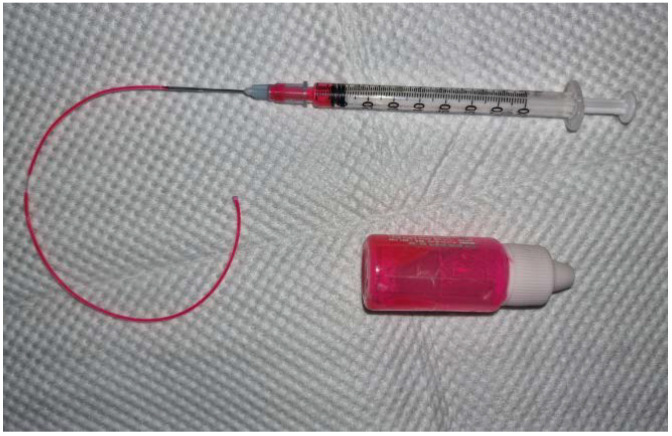
Tattoo ink and a customized instrument for ink placement. The customized instrument is a PTFE Teflon tube with 0.5 mm inner diameter and 0.9 mm outer diameter attached to a 1 mm syringe.

**Figure 2 vetsci-09-00069-f002:**
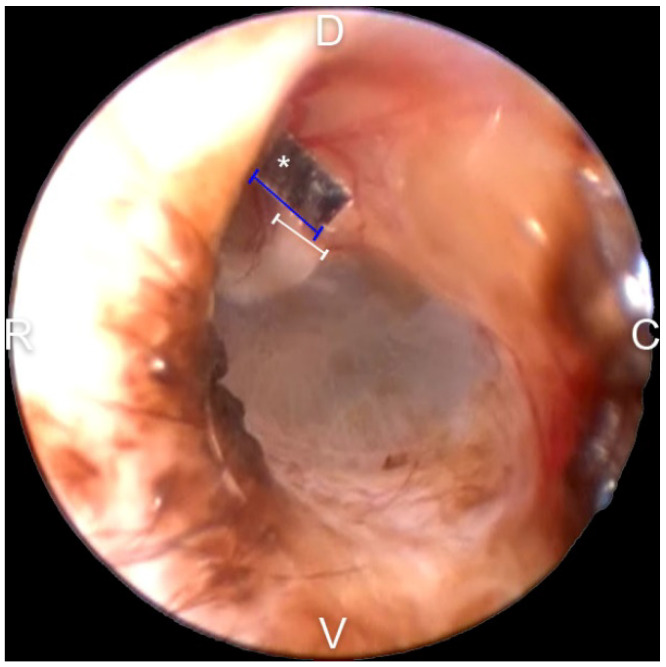
Malleus width measurement with 1.5 × 1.5 mm^2^ tailored 3M tape as a referential scale. The referential scale was placed near the short process of the malleus. Then, using Adobe Photoshop, the width of the scale and the short process of the malleus were measured in pixels. Then, using the ratio, the actual size was measured indirectly: asterisk—referential scale; blue line—width of the referential scale; white line—width of the malleus; D—dorsal; V—ventral; C—caudal; and R—rostral.

**Figure 3 vetsci-09-00069-f003:**
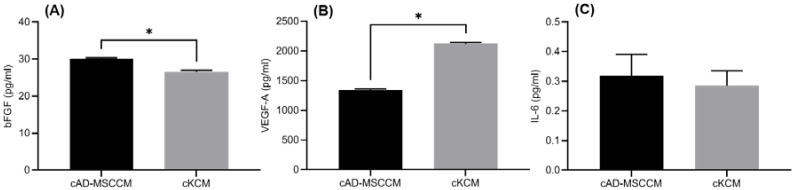
Comparison of basic fibroblast growth factor (bFGF): (**A**), vascular endothelial growth factor-A (VEGF-A) (**B**), and interleukin-6 (IL-6) (**C**), concentrations between canine adipose tissue-derived mesenchymal stem-cell-conditioned media (cAD-MSCCM) and canine keratinocyte-conditioned media (cKCM) (*: *p* < 0.05).

**Figure 4 vetsci-09-00069-f004:**
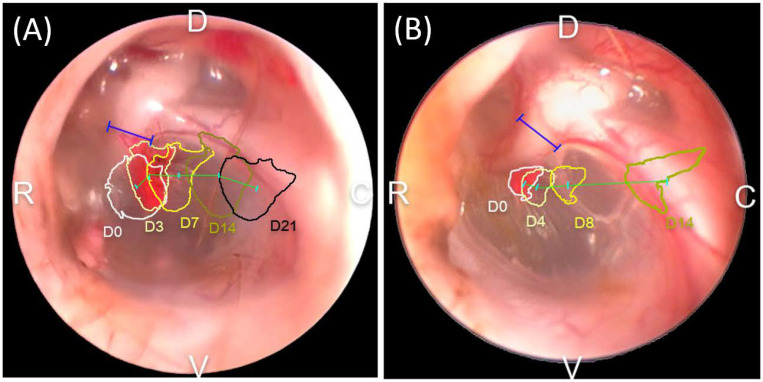
Gross comparison of the tympanic membrane (TM) epithelial migration (EM) rate in the control group (**A**) and treatment group (**B**). The ink drop path over time in the treatment group is visualized in progressively darker colors. The ink drop deviated faster in the treatment group than in the control group and hence had a shorter evaluation period: blue line—width of the malleus; cyan line—midpoint of the ink drops; green line—migrated distance between the two chronological time intervals; D—dorsal; V—ventral; C—caudal; and R—rostral.

**Figure 5 vetsci-09-00069-f005:**
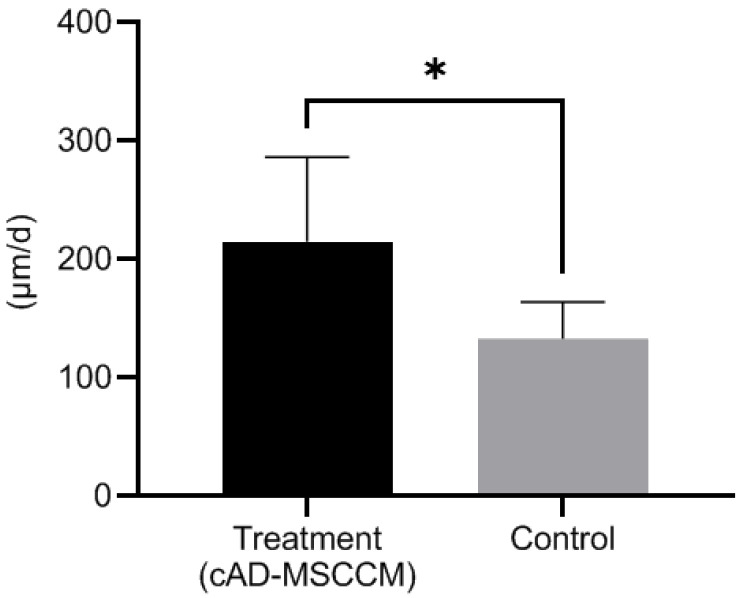
Comparison of TM EM rate in the treatment and the control groups. The TM EM rate was significantly higher in the treatment group than in the control group according to the paired *t*-test (*: *p* < 0.05).

**Table 1 vetsci-09-00069-t001:** Width of malleus measured utilizing a reference scale and digital imaging program.

	Width of the Malleus (μm)	
	Left Ear (LE)	Right Ear (RE)
Beagle 1	1129.4	957.3
Beagle 2	1146.5	958.0
Beagle 3	1217.9	1011.3
Beagle 4	763.5	993.5
Total	1064.0 ± 204.2	980.0 ± 26.8
	1022 ± 142.2	

There was no significant difference in the width of the malleus between LEs and REs (*p* > 0.05).

**Table 2 vetsci-09-00069-t002:** Average TM EM rate per day was calculated for LEs and REs.

	TM EM Rate per Day (μm/d)	
Side of Ear	Treatment Group	Control Group
LE	208.7 ± 62.1	128.8 ± 26.0
RE	216.3 ± 79.9	136.3 ± 37.8
Total	212.5 ± 66.4	133.1 ± 30.9

Trauma: The ear was damaged, and its value was excluded. No significant difference between the LE and RE from the treatment group (*p* > 0.05). No significant difference between the LE and RE from the control group (*p* > 0.05).

## Data Availability

All data generated or analysed during this study are included in this published article. Further enquiries can be directed to the corresponding authors.

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
