# Peer review of "Canine Stem Cell Conditioned Media Accelerates Epithelial Migration in the Canine Tympanic Membrane"

_vetsci, 2022, doi:10.3390/vetsci9020069_

Round 1

Reviewer 1 Report

cAD-MSCCM application / treatment is not clearly explained in materials & methods - some explanations are needed- how many drops exactly (1 or 2) ? How often ? Some details are presented in the abstract but are not clearly exposed in materials and methods.

 I do not totally agree with you when you argue that the in vitro trial elucidated the basis for the phenomenon in vivo. According to me, it is necessary to compare the effect of the application of cKCM and cAD-MSCCM in vivo. Indeed it seems difficult to me to state that the higher level of bFGF in cAD-MSCCM suggests that bFGF is the factor that most strongly stimulates TM EM acceleration.

Reviewer 2 Report

I have read the manuscript with high interest, since a novel and very inovative approach for TM healing acceleration was tested. the idea is brilliant!

Since the manipulation in the ear canal by using videootoscopy is definitively a very challanging and skills demanding method, I am very interested to see the collected data. Can you please provide details (figures)/ table, how the individual calculations measurements on the EM speed were produced?

Regarding the customized instrument for the ink droplet application, would you please provide us with a picture?

How deep did you apply the one or two drops of the treatment fluid in the treated patients? I wonder, if the small amount of the substance could reach the TM?

If I understand properly, you have used the same patients for the control and later for the treatment assessment. So, did you paint the new ink dots to start with the measurments in the treatment group? If so, how long did you wait, between you have used the patient in the traetment group, after you have done the control group with the patient?

Can you provide us with details of MSCCM production? How exactly was the fatty tissue collected? How much?

Figures for references should be normally placed after the comma or dot. E.g.: Ear is an important organ,1   

Ear is an important organ.1
